# *Magnaporthe oryzae* Effector AvrPik-D Targets Rice Rubisco Small Subunit OsRBCS4 to Suppress Immunity

**DOI:** 10.3390/plants13091214

**Published:** 2024-04-27

**Authors:** Linlin Song, Tao Yang, Xinxiao Wang, Wenyu Ye, Guodong Lu

**Affiliations:** 1State Key Laboratory of Ecological Pest Control for Fujian and Taiwan Crops, College of Plant Protection, Fujian Agriculture and Forestry University, Fuzhou 350002, China; llsong2332@163.com (L.S.); yatao8106@163.com (T.Y.); 15806623113@163.com (X.W.); 2China National Engineering Research Center of JUNCAO Technology, Fujian Agriculture and Forestry University, Fuzhou 350002, China

**Keywords:** rice, Rubisco, AvrPik-D, *Magnaporthe oryzae*, resistance, OsRBCS4

## Abstract

Rice blast, caused by the fungal pathogen *Magnaporthe oryzae* (*M. oryzae*), is a highly destructive disease that significantly impacts rice yield and quality. During the infection, *M. oryzae* secretes effector proteins to subvert the host immune response. However, the interaction between the effector protein AvrPik-D and its target proteins in rice, and the mechanism by which AvrPik-D exacerbates disease severity to facilitate infection, remains poorly understood. In this study, we found that the *M. oryzae* effector AvrPik-D interacts with the Rubisco (ribulose-1,5-bisphosphate carboxylase/oxygenase) small subunit OsRBCS4. The overexpression of the *OsRBCS4* gene in transgenic rice not only enhances resistance to *M. oryzae* but also induces more reactive oxygen species following chitin treatment. OsRBCS4 localizes to chloroplasts and co-localizes with AvrPik-D within these organelles. AvrPik-D suppresses the transcriptional expression of OsRBCS4 and inhibits Rubisco activity in rice. In conclusion, our results demonstrate that the *M. oryzae* effector AvrPik-D targets the Rubisco small subunit OsRBCS4 and inhibits its carboxylase and oxygenase activity, thereby suppressing rice innate immunity to facilitate infection. This provides a novel mechanism for the *M. oryzae* effector to subvert the host immunity to promote infection.

## 1. Introduction

Rice (*Oryza sativa*) is a vital food crop, nourishing over half of the global population. Rice blast, caused by the fungus *Magnaporthe oryzae*, is one of the most destructive diseases affecting cultivated rice, posing an ongoing threat to global food security [1,2]. Plants have evolved two tiers of defense against pathogens over the course of their longstanding battle. One tier is the innate immune response, known as PAMP-triggered immunity (PTI), which is activated by pathogen-associated molecular patterns (PAMPs); the other is effector-triggered immunity (ETI), initiated by pathogen effectors [3,4,5]. PAMPs on the surface of pathogens are recognized by pattern recognition receptors (PRRs) on the surface of plant cells, thereby activating PTI. The activation of PTI promptly results in the generation of reactive oxygen species (ROS) within plant tissues, thereby increasing the host resistance to invading pathogens [5]. Owing to their long-term co-evolution, certain pathogens have devised strategies to circumvent host PTI immunity. These pathogens inject effector molecules into host cells to suppress the PTI response [6]. In response, host cells activate a secondary immune system, known as ETI, which involves resistance (R) proteins recognizing the effectors. ETI responses typically culminate in hypersensitive reactions (HRs), conferring elevated resistance levels against pathogens [7]. However, the two principal plant immune pathways, PTI and ETI, are not independent; instead, they reinforce and synergize with each other. The absence of the initial PTI system significantly diminishes the disease resistance conferred by the subsequent ETI system layer. The ETI system in plants can also enhance the PTI by upregulating core protein components within the PTI system [8].

*Magnaporthe oryzae* is an airborne pathogenic fungus that causes rice blast disease. It is highly adaptable and can infect rice at all stages of development [9]. The adaptability of *M. oryzae* is mainly caused by mutations, leading to the instability of the avirulence gene (*AVR*), thereby overcoming the resistance mediated by the main R gene [10]. Recognition is usually mediated by gene-for-gene interactions between host plants and pathogens [11]. In this classic genetic model, the match between a single plant disease resistance gene (R) and a single pathogen avirulence effector gene Avr leads to pathogen recognition and induces plant immunity [11]. The potential molecular mechanism of Pik’s allele recognition of AvrPik variants has been extensively studied [12]. In rice and rice blast populations, both Pik-1/Pik2 NLR gene pairs and AvrPik genes are present in allelic sequences. The Pik locus is located on chromosome 11 of rice [13,14,15,16] and 9 genes have been identified, namely *Pik**, *Pikm*, *Pikp*, *Pikh*, *Pi1*, *Pi7*, *Piks*, *Pik* (*Pik* NLs), and *Pike* [12,13,15,17,18,19,20,21,22,23]. Six Avrpik variants have been identified, namely *AvrPik-A*, *AvrPik-B*, *AvrPik-C*, *AvrPik-D* (also known as *AvrPik-H*), *AvrPik-E*, and *AvrPik-F*, with 1–3 amino acid differences between these variants [22,24,25,26]. The results of the evolutionary analysis show that among *AvrPik-A*, *AvrPik-B*, *AvrPik-C*, *AvrPik-D*, and *AvrPik-E*, *AvrPik-D* is the most abundant and also the most primitive in evolution. The other four alleles have evolved from *AvrPik-D*, and the driving force of evolution is the selective pressure exerted by direct interaction with the corresponding disease resistance protein Pik in rice [22]. *Pikp* only recognizes *AvrPik-D*, while Pikm recognizes *AvrPik-D*, *AvrPik-A*, and *AvrPik-E* [22,27]. Based on the specificity of different effector recognition, the difference between *Pikp-1* and *Pikm-1* lies in the different HMA domain sequences.

Naturally, the Avr proteins act as virulence agents to host plants lacking resistance genes [3]. However, the virulence functions of the *M. oryzae* Avr protein are largely uncharacterized. Emerging evidence indicates that Avr proteins manipulate immunity by targeting various host factors. For example, AvrPiz-t suppresses the E3 ligase activity of RING-type ubiquitin E3 ligases APIP6 and APIP10, facilitating their degradation and thus altering host immunity [28,29]. AvrPiz-t also targets the bZIP transcription factor APIP5, mitigating APIP5-induced cell death and encouraging tissue necrosis [30]. AvrPi9 destabilizes OsRGLG5 to impede basal resistance against *M. oryzae*; conversely, OsRGLG5 ubiquitinates and degrades AvrPi9 [31]. Furthermore, AvrPi9 interaction with ANIP1 differentially modulates rice immunity, contingent on the presence or absence of specific resistance proteins [32]. AvrPita undermines host immunity through interaction with OsCOX11, a cytochrome c oxidase assembly protein, thus inhibiting ROS accumulation and compromising rice immunity [33]. Our prior research demonstrated that AvrPik-D, targeting the transcription factor WG7, enhances its transcriptional activity to undermine rice innate immunity [34].

Molecular mechanisms underlying the recognition of *M. oryzae* AvrPik variants by rice Pik alleles have been extensively characterized [12]. Meanwhile, as a virulence factor, how AvrPik overcomes rice immunity to facilitate *M. oryzae* infection remains poorly understood. In this work, a Rubisco small subunit named OsRBCS4 is found to be a target of AvrPik-D. The overexpression of the *OsRBCS4* gene reduces infection by *M. oryzae*. AvrPik-D localizes to chloroplasts to inhibit the carboxylase and oxygenase activity of Rubisco to suppress rice innate immunity.

## 2. Results

### 2.1. AvrPik-D Interacts with the OsRBCS4 In Vitro and In Vivo

To elucidate the molecular mechanism underlying the suppression of host PTI mediated by AvrPik-D, we use AvrPik-D_22-113_ (which lacks the N-terminal signal peptide, ΔSP) as bait to screen a blast-fungus-infected rice cDNA library. A small subunit of Rubisco OsRBCS4 (LOC_12g19470) was identified. In rice, the OsRBCS4 cDNA contains a 528 bp open reading frame that encodes a protein of 176 amino acids. The one-by-one yeast two-hybrid (Y2H) results confirmed the interaction between AvrPik-D and the full-length coding sequences of OsRBCS4 (Figure 1A). To further verify the direct interaction between the AvrPik-D and OsRBCS4, we expressed AvrPik-D protein tagged with glutathione S-transferase (GST) and OsRBCS4 protein tagged with MBP, respectively, and then performed a GST pull-down assay using GST-AvrPik-D protein as the bait. The result showed that GST-AvrPik-D was captured with MBP-OsRBCS4 protein, indicating that AvrPik-D directly interacts with rice OsRBCS4 (Figure 1B). To test if the interaction also occurs in vivo, we used the LCI and co-immunoprecipitation (CoIP) method. In the LCI assay, fluorescent signals from the interaction between AvrPik-D and OsRBCS4 were detected at the site where AvrPik-D-nluc and cluc-RBCS4 were co-injected (Figure 1C). In the CoIP assay, the plasmids of AvrPikD-GFP (AvrPik-D (ΔSP) fused with the C-terminal GFP tag) and OsRBCS4-HA (OsRBCS4 fused with the C-terminal HA tag) were either co-expressed or separately expressed in Nicotiana benthamiana leaves using the agroinfiltration method. The AvrPikD-GFP fusion protein was discerned in the immunocomplex of OsRBCS4-HA with the anti-GFP antibody (Figure 1D) when anti-GFP IgG beads were employed to immunoprecipitate the AvrPikD-GFP fusion protein from the flora extract. However, no visible signal was detected in the samples expressing only AvrPikD-GFP. Collectively, these results suggest that the AvrPik-D interacts with OsRBCS4 in vivo and in vitro.

### 2.2. OsRBCS4 Is Co-Localized with AvrPik-D in the Chloroplasts

The subcellular localization of OsRBCS4 was explored by fusing the protein with green fluorescent protein (GFP) and expressing it within rice protoplasts, directed by the maize ubiquitin promoter. Confocal microscopy showed that OsRBCS4-GFP was located in the point structure of rice protoplasts compared with the GFP protein (Figure 2A), and the signal of GFP and chloroplast autofluorescence had co-localized in the chloroplasts (Figure 2B). This finding suggests that OsRBCS4 localize to the chloroplasts in rice. Our previous research showed that M. oryzae effector AvrPik-D localizes in the cytoplasm and nucleus [34]. The subcellular localization of OsRBCS4 prompted us to determine whether AvrPik-D is co-localized with OsRBCS4 in rice cells. Upon co-expressing OsRBCS4-GFP with RFP-AvrPik-D in rice protoplasts, we observed overlapping signals of OsRBCS4-GFP and RFP-AvrPik-D within the chloroplasts (Figure 2C). These results suggest that the AvrPik-D may go into plant chloroplasts to target OsRBCS4 to promote fungal colonization.

### 2.3. Overexpression of OsRBCS4 Enhances Resistance to M. oryzae

To elucidate the role of OsRBCS4 in rice immunity against M. oryzae, OsRBCS4-overexpressing (OE) transgenic lines were generated in the Minghui86 (MH86) background, utilizing the maize ubiquitin promoter for constitutive expression. Three OsRBCS4 overexpression transgenic lines, identified by qRT-PCR (Appendix A) and designated as OsRBCS4-OE-1, OsRBCS4-OE-2, and OsRBCS4-OE-3, were selected for molecular and phenotypic analyses. We performed a punch inoculation and measured the basal infection levels of the transgenic plants. After punch inoculation, we found that the OsRBCS4-OE mutants developed smaller lesions and exhibited lower relative fungal biomass than the wild type (WT) plants (Figure 3A–C). To elucidate the mechanism of OsRBCS4-mediated rice resistance, the expression levels of defense-related genes in OsRBCS4-OE lines were measured using qRT-PCR. The analysis showed that the expressions of pathogenesis-related (PR) genes, OsPR1b [35], OsPAL6 [36], and OsPBZ1 [37], were significantly upregulated in OsRBCS4-OE lines compared to in WT plants (Figure 3D–F). These results indicate that the OsRBCS4-OE mutants exhibited less infection by *M. oryzae* than the WT plants.

### 2.4. Overexpression of OsRBCS4 Enhances PAMP-Triggered ROS Production in Rice Leaves

To further explore how OsRBCS4 positively regulates rice resistance, ROS generation was measured in OsRBCS4-overexpressed plants following chitin treatments. PAMP elicitors activate OsCEBiP and OsCERK1-mediated PTI signaling in rice [38]. The overexpression of OsRBCS4 led to higher ROS production relative to the WT plants (MH86) after chitin treatment. It was observed that ROS levels peaked 16 min post-chitin treatment and were significantly higher in OsRBCS4-OE mutants than in WT plants (Figure 4A). Moreover, the maximum ROS value in the OsRBCS4-OE mutants was also higher (Figure 4B). These results indicate that the overexpression of OsRBCS4 in rice enhances the chitin-induced ROS burst, further confirming the role of OsRBCS4 in the activation of the PTI response in rice. In addition, during the rice blast infection, OsRBCS4-OE lines generate more H_2_O_2_ (DAB staining) around fungal appressoria than WT plants (Figure 4C), consistent with their higher resistance.

### 2.5. AvrPik-D Weakens the Activity of Rubisco

The chloroplasts co-localization of AvrPik-D and OsRBCS4 refer us to explore their roles in Rubisco activity regulation. We tested the Rubisco activity in OsRBCS4-overexpression plants, and the result shows that the activity was significantly increased compared to wild-type rice MH86 (Figure 5A). This finding suggests that the Rubisco activity may influence plant resistance to pathogens.

To elucidate the relationship between AvrPik-D and OsRBCS4, the expression levels of OsRBCS4 were assessed in AvrPik-D-overexpressing transgenic rice via qRT-PCR. The results indicated that OsRBCS4 transcript levels were significantly reduced in AvrPik-D transgenic rice compared to NPB (Figure 5B). This result suggests that AvrPik-D may inhibit OsRBCS4 at the transcriptional level. To explore AvrPik-D impact on rice Rubisco activity, a crude enzyme was extracted from wild-type plants, and changes in Rubisco activity were assessed following the addition of purified AvrPik-D protein. The result shows that the total Rubisco activity significantly decreased following the increase in AvrPik-D compared to the control (Figure 5C). The above results suggest that AvrPik-D inhibits the Rubisco activity to suppress plant immunity.

## 3. Discussion

Rubisco (ribulose-1,5-bisphosphate carboxylase/oxygenase) is the major enzyme assimilating CO_2_ into the biosphere [39]. The holoenzyme of Rubisco includes eight large subunits (RBCLs) and eight small subunits (RBCSs), that is, an L_8_S_8_ structure [18], in most chemoautotrophic bacteria, cyanobacteria, and algae, and all land plants. Rubisco small subunits (RBCSs) are encoded by a nuclear multigene family in plants. In rice [40,41], five RBCS genes (*OsRBCS1*, *OsRBCS2*, *OsRBCS3*, *OsRBCS4*, and *OsRBCS5*) have been identified. Although Rubisco is a critical enzyme in photosynthesis, its role in plant immunity is largely unknown. Only a few reports showed that the small subunits of Rubisco are involved in plant immune regulation. A clear case demonstrates in *Nicotiana benthamiana* that the silencing of NbRbCS allowed the Tomato Mosaic Virus (ToMV) to provoke leaf necrosis, thereby enhancing local viral infectivity. However, a delay was observed in the development of systemic viral symptoms. The aforementioned findings suggest that NbRbCS plays a crucial role in ToMV movement and the mechanisms of plant antiviral defenses [42]. Recently, Qin et al. reported that potyvirids have co-opted NbRbCS, using it as a scaffold protein for assembling a complex for viral intercellular movement [43]. In this study, we found that AvrPik-D interacts with the Rubisco small subunit OsRBCS4 in vitro and in vivo, and *OsRBCS4* positively regulates rice resistance to *M. oryzae*.

Chitin, a typical component of the fungal cell wall, can induce PTI in plants [44]. Our results showed that *OsRBCS4*-overexpression plants accumulate ROS to a higher level than wild-type plants after elicitor treatments, indicating that *OsRBCS4* functions as a promoter of ROS accumulation in rice. Additionally, the expression analysis indicated that the SA pathway marker genes *OsPR1b*, *OsPAL6*, and *OsPBZ1* were significantly upregulated in *OsRBCS4*-OE plants. These findings suggest that *OsRBCS4* activates the PTI response through the modulation of the SA-mediated defense pathway.

Under normal conditions, chloroplasts are the center for carbon fixation and nitrogen assimilation. However, during ETI, chloroplasts undergo reprogramming to become centers for ROS production, defense-related hormones, and metabolites [45,46,47,48]. Rubisco-mediated oxygenation consumes O_2_, and low O_2_ levels could also minimize chloroplastic ROS (cROS) production [49]. The accumulation of cROS has been observed in PTI, ETI, and various host–microbe interactions [46,47,48,50,51,52,53,54,55]. In our study, OsRBCS4 was found to localize within rice chloroplasts and enhanced the levels of ROS in the *OsRBCS4*-OE lines (Figure 4). These results suggest that OsRBCS4 may enhance plant immunity through facilitating ROS production. Previous studies have found that AvrPik-D localizes in the plant nucleus and cytoplasm [34] and AvrPik-D inhibits ROS accumulation and suppresses innate immunity in rice [56]. However, we found that AvrPik-D and OsRBCS4 were co-expressed in rice chloroplasts. These results lead us to hypothesize that AvrPik-D might be transferred to chloroplasts to inhibit ROS production and suppress plant immunity.

Under antisense conditions, mRNA levels become limiting for Rubisco synthesis [57]. For example, the antisense suppression of RBCS reduces the Rubisco content in several plant species including tobacco [58,59], Flaveria [60], and rice [61]. In this study, the overexpression of *OsRBCS4* enhances rice Rubisco activity, and the AvrPik-D inhibited Rubisco activity in vitro. Therefore, we hypothesize that when *M. oryzae* infects rice, the AvrPik-D binds to the target OsRBCS4, thereby inhibiting its Rubisco activity and affecting rice resistance to promote infection. However, the transcription level of *OsRBCS4* was also inhibited by AvrPik-D. Further research is needed to understand this transcriptional regulation of AvrPik-D on *OsRBCS4*. Interestingly, the *M. oryzae* effector AvrPik-D can also target and promote the transcriptional activity of the negative regulatory gene WG7, thereby inhibiting rice innate immunity and promoting infection [34]. This is impressive as the rice blast fungus regulates target genes through different strategies to achieve the goal of infecting rice.

## 4. Materials and Methods

### 4.1. Plant Materials and Growth Conditions

Rice seeds of Minghui86 (MH86), OsRBCS4-OE (MH86 background), and AvrPik-D-overexpressing transgenic rice (Nipponbare background) were placed at 37 °C for germination, and then the seeds were grown in a climate greenhouse at 28 °C and 70% relative humidity and in 12 h light/12 h darkness. *N. benthamiana* was planted at 24 °C with 16 h of light/8 h of darkness.

### 4.2. Yeast Two-Hybrid (Y2H) Assay

The coding region of *AvrPik-D* (lacking the N-terminal signal peptide, ΔSP) was cloned into the pGBKT7 (BD) vector in our previous research [34]. The full-length CDS of the *OsRBCS4* genes was cloned into the pGADT7 vector and then the corresponding plasmids were co-transformed into AH109 cells following the manufacturer’s instructions. The preparation and transformation of yeast receptive cells are based on Matchmaker ^TM^ Gold Yeast Two Hybrid System User Manual.

### 4.3. Pull-Down Assay

This assay was performed as described by Han et al. [62]. Briefly, the CDS of *AvrPik-D* and *OsRBCS4* was individually subcloned into pGEX-4T-1 (containing N-GST tag) and the prokaryotic expression vector pMAL-c2x (containing N-MBP tag). The GST-AvrPik-D and MBP-OsRBCS4 plasmids used for the pull-down assay were generated. The fusion proteins were expressed in *E. coli* BL21 cells. The cells were collected and resuspended in a lysis buffer (50 mM PBS, 0.1 mM PMSF, and 1 × protease inhibitor cocktail) and sonicated on ice. The mixture was centrifuged at 10,000× *g* at 4 °C for 30 min. The supernatant was then transferred and incubated with GST beads at 4 °C for 2 h and washed with the lysis buffer. The supernatant containing the expressed MBP-OsRBCS4 protein was added to GST-AvrPik-D-protein-precipitated GST beads and incubated at 4 °C for another 4 h with gentle end-over-end mixing. MBP was a negative control. After six washes with the lysis buffer, the beads were collected for SDS-PAGE and Western blot analyses. Specific anti-GST and anti-MBP antibodies were used for the immunoblot analysis.

### 4.4. Co-IP Assay

This assay was performed as described by Yang et al. [34]. The CDS of *OsRBCS4* (with HA tag) and *AvrPik-D* (ΔSP, with GFP tag) was cloned into pVX and pCXSN vectors, driven by the 35 S promoter. The pVX-RBCS4-HA and pCXSN-AvrPikD-GFP vectors were, respectively, transformed into agrobacterium tumefaciens GV3101, and then co-injected into tobacco leaves. The infiltrated leaves were harvested after 60–72 h and then ground to powder in liquid nitrogen. The powder was suspended in the lysis buffer (10 mM Tris-HCL, pH 7.5, 0.5 mM EDTA, 150 mM NaCl, 0.5% NP40, 1 mM PMSF, and 1 x protease inhibitor cocktail). Anti-GFP agarose bead suspension (M20015, Absmart, Shanghai, China) was added to the protein lysates. The mixture was incubated at 4 °C for 8 h with constant end-over-end rotation. The beads were then rinsed with a wash buffer until the A280 of the supernatant fraction was less than 0.01. Anti-GFP agarose mixtures were collected for a Western blot and SDS-PAGE analysis. OsRBCS4-HA and AvrPikD-GFP fusion proteins were detected by anti-HA and anti-GFP antibodies, respectively, according to the manufacturer’s instructions (M20003 or MA9023, Abmart, Shanghai, China).

### 4.5. Split-LUC Assay

The CDS sequences of *OsRBCS4* and *AvrPik-D* were inserted into pCAMBIA-35S-nLuc and pCAMBIA-35S-cLuc, respectively. The split-LUC assays were performed as previously described [63]. In brief, the plasmids were transformed into *Agrobacterium* GV3101. After a culture in LB media overnight at 28 °C, the bacteria were collected and resuspended in an infiltration buffer (10 mM MgCl_2_, 10 mM MES, 200 mM acetosyringone, pH 5.7), and incubated for 2–3 h at 28 °C. The suspensions were injected (A_600nm_ = 0.5) into leaves of four-week-old *N. benthamiana* using a needleless syringe. The infiltrated plants were kept at room temperature for 48–72 h and were then spayed with 1 mM luciferin, and the leaves were kept in the dark for 5–10 min to quench the fluorescence. A cooling CCD imaging apparatus was used for image capture.

### 4.6. Subcellular Localization

The full-length coding regions of *OsRBCS4* were amplified using the appropriate primers (Appendix A) and inserted into a pHF223 vector (Ubiquitin1 (Ubi1) promoter) with in-frame GFP fusion. *Avr-Pik-D* (Δ*SP*) was inserted into the pRTVnRFP vector (Ubi1 promoter) with in-frame RFP fusion. To observe the protein localization, rice protoplasts were prepared and plasmid transformation was performed according to the methods of previous research [30,64,65]. The transformed protoplasts were observed by fluorescence confocal microscopy (Nikon A1, Tokyo, Japan). The fusion protein was visualized at 488 nm excitation and 561 nm excitation, respectively.

### 4.7. RNA Isolation and Transcript Analysis

Following the guidelines laid by the manufacturer, total RNA was derived from the tissue of rice leaves via the Eastep^®^ Super Total RNA Extraction Kit (LS1040, Promega Shanghai, Shanghai, China). Utilizing Evo M-MLV Mix Kit with gDNA Clean for qPCR (AG11705, Accurate Biology, Changsha, China), 1 μg of total RNA was reverse-transcribed into the first-strand cDNA. The manufacturer’s protocol was followed to quantify transcript levels by qRT-PCR, deploying the SYBR Green Premix Pro Tag HS qPCR Kit (AG11701, Accurate Biology, Changsha, China). The relative quantification of the target gene expression level was normalized to that of *OsActin* using the 2^−ΔΔCt^ method [66].

### 4.8. Construction of Transgenic Plants

The construction of an *OsRBCS4*-overexpression vector involved the amplification of the CDS of *OsRBCS4* from rice cDNA using the primers OE-F and OE-R (Appendix A), which was then integrated into the binary pCAMBIA1301 plasmid. The plasmid thereafter underwent transformation into the *Agrobacterium tumefaciens* strain EHA105 and was later applied in *Agrobacterium*-mediated genetic transformation. Transgenic rice plants were generated in the MH86 background and rice transformation was performed by Wuhan BioRun Biosciences Co., Ltd. (Wuhan, China).

### 4.9. Pathogen Inoculation Assay

The *M. oryzae* isolates 501-3 were cultured on a complete medium in darkness at 28 °C for 1 week; subsequently, the mycelium was chopped into small pieces and moved to a rice bran culture medium for sporulation. The punch inoculation proceeded according to previous research methods of Park et al. [28] with a slight modification. Leaves of plants, aged between 6 and 8 weeks, were gently punctured and treated with a 7 μL spore suspension (5 × 10^5^ spores/mL). Lesion diameters were measured at 10 days after inoculation using ImageJ software. The quantification of the fungal biomass in the infected rice leaf was performed using a previously described method [28]. In brief, the DNA of rice leaves (about 4 × 1 cm) was extracted by the CTAB method. DNA-based qPCR was performed and relative fungal growth was calculated using the threshold cycle value (C_T_) of *M. oryzae* Pot2 DNA against the C_T_ of rice genomic *ubiquitin* DNA (OsUG). Relative fungal growth was calculated by the equation 2^CT(OsUG)−CT(MoPot2)^.

### 4.10. Measurement of ROS and DAB Staining

The luminol-based methodology was slightly altered from a previous description [28] to perform ROS measurement. A summarized process is as follows: Leaf sheaths from 10-day-old rice plants grown on a 1/2 MS medium were gathered (approximately 4 mm strips) and pre-incubated overnight in sterilized distilled water. Subsequently, these sheaths were transferred to a 96-well microplate containing a reaction buffer with 20 mM luminol (120-04891, FUJIFILM Wako, Osaka, Japan) and 5 mg/mL of horseradish peroxidase (P8375, Sigma, St. Louis, MO, USA). After standing still for 30 min, chitin (8 nM) was rapidly added. Water was used as a control. Luminescence was monitored immediately after the treatment and was continuously measured at 1 min intervals for 50 min using a Varioskan Flash multireader (Thermo Fisher Scientific, Waltham, MA, USA).

The DAB-stained sections were observed under a microscope. For rice cell DAB staining purposes, the leaf sheath of rice was exposed to the *M. oryzae* isolate 501-3. Previous descriptions [67] guided the monitoring of H_2_O_2_ accumulation via DAB staining. In darkness at 22 °C, the leaf sheath sections were incubated, and placed in 1 mg/mL of DAB for a 10 h period. Observations of the DAB-stained sections were conducted under a microscope.

### 4.11. Measurements of Leaf Enzymes’ Activities

Rubisco activity assays were conducted as described by Liang, et al. [67]. Using a plant Rubisco ELISA kit from Shanghai Enzyme-linked Biotechnology Co., Ltd., Shanghai, China, activity of Rubisco was determined, adhering strictly to the procedure provided by the manufacturer. Grind the rice leaves in liquid nitrogen, take 1.0 g of powder, and transfer to a 10 mL tube. This tube contains 2 mL of a natural extraction buffer (containing 50 mM Tris MES, pH 8.0, 0.5 M sucrose, 1 mM MgCl_2_, 10 mM EDTA, pH 8.0, 5 mM dithiothreitol [DTT], and 1 × protease inhibitor cocktail). This is followed by centrifugation at 10,000× *g* for 10 min to obtain a crude enzyme extract. The microporous plate in this kit is coated with excessively purified plant anti-RubisCO (1,5-diphosphate carboxylase and oxygenase dual antibodies) to produce solid-phase antibodies. Then, the supernatant was added to the microporous plate, and the anti-RubisCO with the HRP label was added to form an antibody–antigen–enzyme-labeled antibody complex. The staining process, which featured a blue color, utilized the TMR substrate. Acid provided in the ELISA kit was added to terminate the reaction. Absorbance at 450 nm was measured spectrophotometrically, and the activities of Rubisco were calculated from the standard curve. The protein content was measured by Pierce^TM^ BCA Protein Assay Kit (23227, Thermo Scientific, USA). In total, 10 μL of the crude enzyme extract was added into 200 μL of a BCA working buffer, and absorbance at 562 nm was measured after incubating at 37 °C for 30 min. The standard curve was constructed by using bovine serum albumin, and the sample’s OD562 value was later logged, after which the corresponding protein content was determined based on the standard curve.

### 4.12. Accession Numbers

The sequence data can be found in Rice Genome Annotation Project (http://rice.uga.edu/index.shtml (accessed on 23 April 2024)). *OsBCS4* (LOC_Os12g19470), *sPR1b* (LOC_Os01g28450), *OsPAL6* (LOC_Os04g43800), *OsPBZ1* (LOC_Os12g36880), *OsActin* (LOC_Os03g50885).

## 5. Conclusions

In this study, we have confirmed the interaction between AvrPik-D and OsRBCS4 using in vivo and in vitro methods. OsRBCS4 is localized in chloroplasts and co-localizes with Avr-Pik-D within the same organelles. This indicates that when the effector protein AvrPikD infects rice, it targets the Rubisco small subunit in chloroplasts. The overexpression of *OsRBCS4* in transgenic rice resulted in the reduced infection of *M. oryzae*. Furthermore, *OsRBCS4* overexpression promoted chitin-induced ROS burst, increased Rubisco activity, and induced defense-related genes (*OsPR1b*, *OsPAL6*, and *OsPBZ1*), leading to enhanced basal resistance against *M. oryzae* infection. The in vitro assay also demonstrated that AvrPik-D inhibited Rubisco activity. Meanwhile, the transcript levels of *OsRBCS4* were downregulated in AvrPik-D transgenic rice. In conclusion, our data strongly support the positive regulatory role of *OsRBCS4* in rice blast resistance. Additionally, we found that AvrPik-D directly interacts with OsRBCS4, leading to the inhibition of Rubisco activity and facilitating *M. oryzae* infection. These significant findings provide valuable insights into the mechanisms underlying *M. oryzae* AvrPik-D’s targeting of rice immunity to promote infection.

## Figures and Tables

**Figure 1 plants-13-01214-f001:**
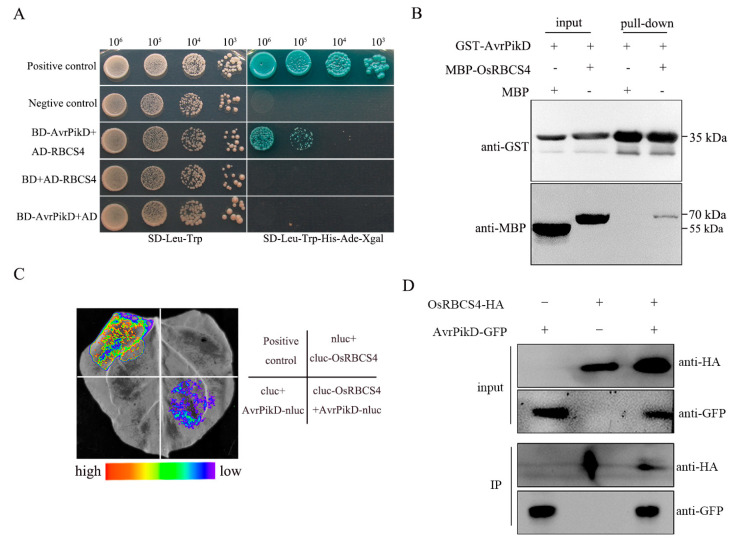
AvrPik-D interacts with OsRBCS4 in vitro and in vivo. (**A**) The Y2H assay tests AvrPik-D interaction with OsRBCS4. Yeast cells containing the specified bait and prey plasmids were diluted 10×, 100×, and 1000×, subsequently getting spotted onto selective media (SD-Leu-Trp and SD-Leu-Trp-His-Ade) with X-α-gal. The applied positive and negative controls were pGADT7-T + pGBKT7-53 and pGADT7-T + pGBKT7-lam, respectively. The selective medium without leucine and tryptophan is indicated by SD-Leu-Trp, while SD-Leu-Trp-His-Ade stands for the selective medium free of leucine, tryptophan, adenine, and histidine. (**B**) Pull-down assays were conducted with fusion proteins (GST:AvrPik-D and MBP:OsRBCS4) using GST beads and evaluated by an immunoblot analysis with anti-GST and anti-MBP antibodies. (**C**) In N. benthamiana leaves, constructs coding for AvrPik-D-nluc (N-terminal segment of LUC) and cluc- OsRBCS4 (C-terminal fragment of LUC) were jointly infiltrated. After three days, the infiltrated leaves were treated with 1 mM luciferin, followed by capturing of the bioluminescence visuals with a CCD camera. The combinations cluc + AvrPik-D-nluc and cluc-OsRBCS4 + nluc were co-infiltrated as the negative controls; cluc-APIP5 + AvrPiz-t -nluc were co-infiltrated as the positive controls. (**D**) The co-immunoprecipitation (Co-IP) assay of AvrPik-GFP and OsRBCS4-HA in *N. benthamiana*. The Co-IP assay was carried out with anti-GFP beads, and the proteins were analyzed by a Western blot with anti-GFP and anti-HA antibodies.

**Figure 2 plants-13-01214-f002:**
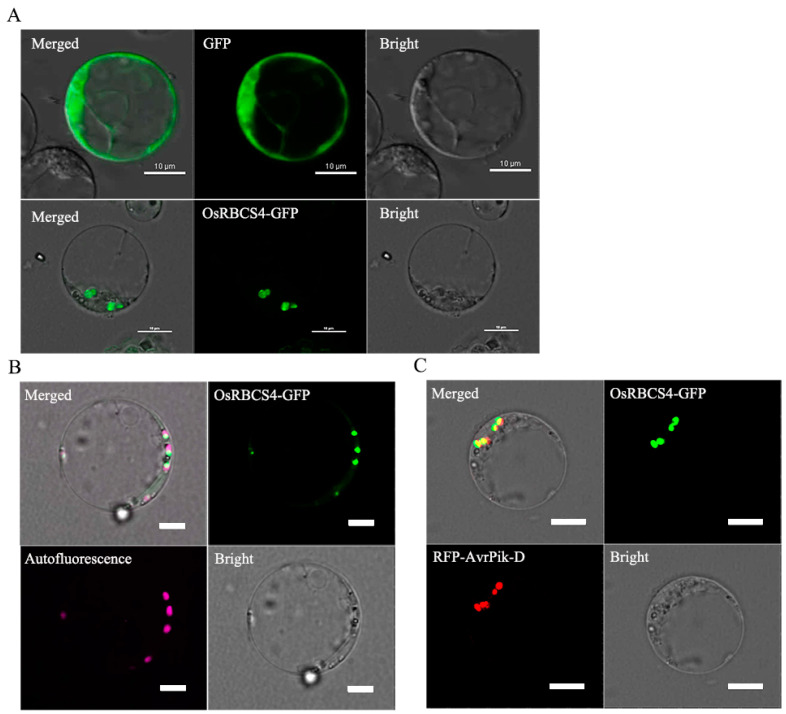
The subcellular localization of OsRBCS4 and its co-localization with AvrPik-D. (**A**) OsRBCS4-GFP was expressed in rice protoplasts. GFP fluorescent signals observed in the point structure of rice protoplasts. Scale bar = 10 μm. (**B**) OsRBCS4-GFP and chloroplast autofluorescence were co-localized in the chloroplast. GFP and chloroplast autofluorescence fluorescent signals overlapped in the chloroplast. The pink fluorescent signal indicated chloroplast autofluorescence. The light green fluorescent signal indicates the co-localization of OsRBCS4 with the chloroplasts. Scale bar = 5 μm. (**C**) GFP-OsRBCS4 and RFP-AvrPik-D were co-expressed in rice protoplasts. GFP and RFP fluorescent signals overlapped in the chloroplast. Scale bar = 10 μm.

**Figure 3 plants-13-01214-f003:**
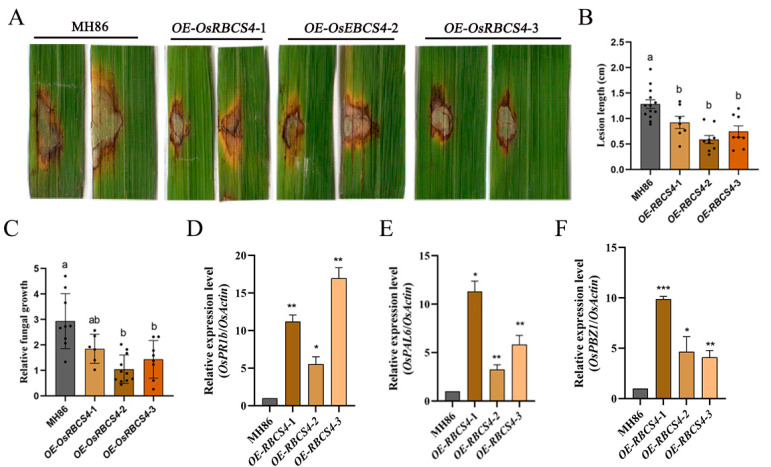
OsRBCS4 positively regulates the resistance of rice against *M. oryzae*. (**A**) Phenotype analyses of *OsRBCS4*-OE and wild type (MH86) plants by *M. oryzae* infection. The punch inoculation of leaves with a conidia suspension of *M. oryzae* isolate 501-3, performed using 6-week-old plants. The images were taken 10 days after inoculation. Experiments were repeated three times with similar results. (**B**) Lesion length was measured using ImageJ v1.46 software. Data are shown as the mean ± SD (different letters indicate *p* < 0.05, n ≥ 7). (**C**) Relative fungal biomass of *M. oryzae* was quantified by qPCR using the fungal gene *MoPot2* and the rice gene *OsUG*. Data are shown as the mean ± SD (different letters indicate a significant difference, *p* < 0.05, n ≥ 6). (**D**–**F**) The analysis of *PR* genes in the leaves of *OsRBCS4* overexpression lines and WT plants. Data were normalized to the expression level of *OsActin*. Data are shown as the mean ± SD (the * indicates a significant difference, * *p* < 0.05, ** *p* < 0.01, *** *p* < 0.001, n = 3).

**Figure 4 plants-13-01214-f004:**
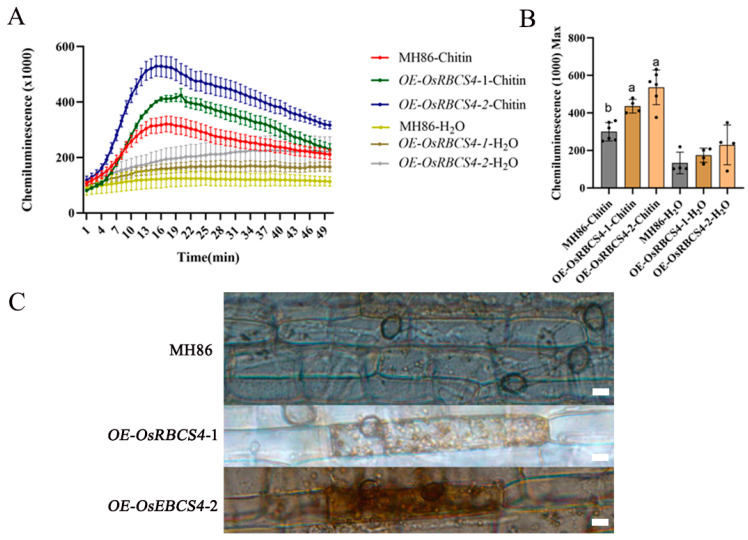
Reactive oxygen species (ROS) and H_2_O_2_ production in *OsRBCS4*-OE and WT plants. (**A**) Chitin-induced ROS burst in *OsRBCS4*-OE and WT plants. Leaf disks were treated with water or 8 nM chitin, and ROS accumulation was measured by a luminol assay. (**B**) Calculate the maximum ROS value for each individual leaf disk. Data are shown as the mean ± SD. (Different letters indicate a significant difference, *p* < 0.05, n ≥ 4.) (**C**) DAB staining showing that *OsRBCS4*-OE lines generate more H_2_O_2_ around appressoria than WT plants. Scale bar = 20 μm.

**Figure 5 plants-13-01214-f005:**
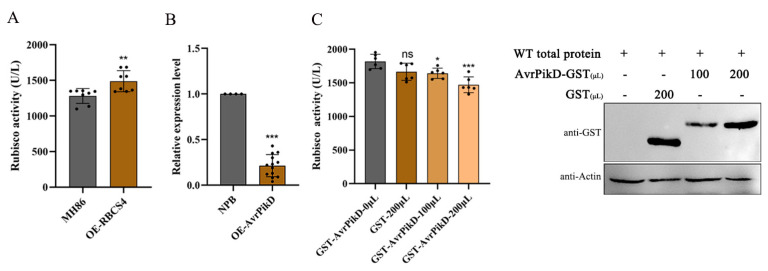
AvrPik-D inhibits Rubisco activity. (**A**) Ribulose 1,5-bisphosphate carboxylase/oxygenase activity in MH86 and *OsRBCS4*-OE plants. Shown are the mean ± SD from three biological replicates each containing three plants. (**B**) The expression level of *OsRBCS4* in NPB and *AvrPikD*-OE plants. Each bar represents the mean standard deviation (SD) (n ≥ 4). Asterisks indicate significant differences analyzed by Student’s *t*-test (* *p* < 0.05, ** *p* < 0.01, *** *p* < 0.0001). (**C**) AvrPik-D inhibits Rubisco activity in vitro (left), and the amount of related protein (right). The purified AvrPikD-GST protein was added to the total protein of wild-type plants’ solution and incubated at 4 °C for 4 h. Activity was estimated using the RubisCO ELISA kit. GST protein as the negative control. Data are shown as the mean ± SD (the * indicates a significant difference, * *p* < 0.05, ** *p* < 0.01, *** *p* < 0.001, n = 6).

## Data Availability

All data generated or analyzed during this study are included in this published article and its supplementary information files.

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
