# Peer review of "Magnaporthe oryzae Effector AvrPik-D Targets Rice Rubisco Small Subunit OsRBCS4 to Suppress Immunity"

_plants, 2024, doi:10.3390/plants13091214_

Round 1
Reviewer 1 Report
Comments and Suggestions for Authors
Dear Editor-in-chief,
This manuscript entitled “Magnaporthe oryzae effector AvrPik-D targets rice Rubisco 2 small subunit OsRBCS4 to suppress immunity” delivers interesting data and excellent interpretation. Therefore, this manuscript can be acceptable to Plants, but there will be small minor revisions required for the final publication. Due to the quantity and quality of these experiment, this manuscript will be an excellent paper published in Plants. The minor revision points are as follow:
1. Line 53: Please add a reference
2. Line 78: Please check the meaning of ToMV is Tomato mosaic virus or Tomato mosaic tobamovirus.
3. Line 117-119: Please revise this part “These results indicate that AvrPik-D interacts with OsRBCS4 in vivo. Collectively, these results suggest that the AvrPik-D interacts with OsRBCS4 in vivo and in vitro”. The author may sum up the sentences.
4. Line 130: N. bethamiana should be in italic
5. Line 144: Please add a full stop after this sentence “Our previous research has shown that M. oryzae effector AvrPik-D localizes in the 144 cytoplasm and nucleus [17]”.
6. Line 156: “Red” should be in small letter.
7. Line 178: Please check 501-3 conidia. Is it correct?
8. Line 183: Fungal gene “MoPot2” should be in italic. Please check through out the manuscript
9. Line 211: Please remove the extra space before “ We tested”
10. Please write the conclusion part in a better way
Again, this manuscript was very well-written based on many excellent experiments, therefore, there will be no point of further recirculation of the revised version of this manuscript (also this manuscript does not need any English editing).
Comments on the Quality of English Language
Dear Editor, This manuscript does not need any English editing.
Author Response
Dear Editor and Reviewers,
Thanks very much for taking your time to review this manuscript. We really appreciate all your comments and suggestions! Please find my itemized responses in below and my revisions/corrections (color with red) in the re-submitted files.
Reviewer 1
This manuscript entitled “Magnaporthe oryzae effector AvrPik-D targets rice Rubisco small subunit OsRBCS4 to suppress immunity” delivers interesting data and excellent interpretation. Therefore, this manuscript can be acceptable to Plants, but there will be small minor revisions required for the final publication. Due to the quantity and quality of these experiment, this manuscript will be an excellent paper published in Plants. The minor revision points are as follow:
- Line 53: Please add a reference
Response: This has been revised in the manuscript. Modify it on line 72 and mark it in red.
- Line 78: Please check the meaning of ToMV is Tomato mosaic virus or Tomato mosaic tobamovirus.
Response: It is Tomato mosaic virus, we revised the manuscript and this paragraph was moved to discussion section according to Reviewer 2 suggestion. Modify it on line 246 and mark it in red.
- Line 117-119: Please revise this part “These results indicate that AvrPik-D interacts with OsRBCS4 in vivo. Collectively, these results suggest that the AvrPik-D interacts with OsRBCS4 in vivo and in vitro”. The author may sum up the sentences.
Response: This has been revised in the manuscript. Modify it on line 119-120 and mark it in red.
- Line 130: N. bethamiana should be in italic
Response: This has been revised in the manuscript. Modify it on line 130 and mark it in red.
- Line 144: Please add a full stop after this sentence “Our previous research has shown that M. oryzae effector AvrPik-D localizes in the cytoplasm and nucleus [17]”.
Response: This has been revised in the manuscript. Modify it on line 147 and mark it in red.
- Line 156: “Red” should be in small letter.
Response: This has been revised in the manuscript. Modify it on line 158 and mark it in red.
7.Line 178: Please check 501-3 conidia. Is it correct?
Response: It is correct, but we modified the sentences to be better understandable. Modify it on line 180 and mark it in red.
- Line 183: Fungal gene “MoPot2” should be in italic. Please check through out the manuscript
Response: This has been revised in the manuscript. Modify it on line 184 and mark it in red.
- Line 211: Please remove the extra space before “We tested”
Response: This has been revised in the manuscript. Modify it on line 212 and mark it in red.
- Please write the conclusion part in a better way
Response: We have revised the conclusion part. Modify it on line 429-442 and mark it in red.
Reviewer 2 Report
Comments and Suggestions for Authors
The manuscript entitled “Magnaporthe oryzae effector AvrPik-D targets rice Rubisco small subunit OsRBCS4 to suppress immunity” by Linlin Song et al. The study described M. oryzae effector AvrPik-D targets the Rubisco small subunit OsRBCS4 and inhibits its carboxylase and oxygenase activity, thereby suppressing rice innate immunity to facilitate infection. However, I think the following comments needs to be addressed before it can be considered further.
Major comments:
1. Figure 2C: The authors described ‘OsRBCS4 was found to localize within rice chloroplasts.’ In author's previous research, the GFP-AvrPik-D were observed in the nucleus and cytoplasm in rice protoplasts, and the overlapped signals with WG7-RFP in the nucleus. While, in this study, why do they observe overlapping signals only within the chloroplasts, not in the nucleus and cytoplasm?
2. Figure 4: The authors described “Three OsRBCS4 overexpression transgenic lines identified by qRT-PCR were selected for molecular and phenotypic analyses.” why ROS generation was measured in OE-OsRBCS4-1and OE-OsRBCS4-2 following chitin treatments? Why tested the Rubisco activity in OE-OsRBCS4? I think the authors need to test the ROS generation and Rubisco activity in the 2-3 independent lines.
Minor comments:
1. Line 21: The significance of this study is missing.
Reviewer 3 Report
Comments and Suggestions for Authors
Rice blast, caused by Magnaporthe oryzae, is a destructive disease of rice, threatening world food security. A series of experiments have been conducted in this research, and the results indicated that M. oryzae effector AvrPik-D interacts with Rubisco small unit OSRBCS4, overexpress OSRBCS4 could enhance ROS production and reduce infection. The findings could help understand the mechanisms of infection-defense interactions of this important disease. This manuscript may be publishable after revisions.
Major concerns
1 .Introduction didn’t sufficiently introduce the status of research on this subject. More details are needed, such as multiple alleles of rice Pik gene and variants of M. oryzae AvrPik have been identified. The Rubisco paragraph could be moved to “Discussion”. There is no need to describe the results in the Introduction part.
2. Half of the discussion was a summary of the results, which is redundant and unnecessary since there is a Conclusion part. Different targets of M. oryzae Avr genes were identified, and the authors’ previous research found that AvrPik-D targets transcription factor WG7. The differences should be discussed.
3. The authors can cite their own paper or re-organize the language, especially the Materials and Methods part.
Minor suggestions
L27 delete the 1st “global’
L45-46 delete “layer”
L88 “enhances resistance” should be replaced with “reduces infection”
L96 change “infection” to ‘infected”
L95-96 and L99-100, redundant and confusing.
L103 change “respectively. And” to “respectively, and”
L106 change "To confirm that” to “To test if”
L107 change "fluorescence” to “fluorescent”
L166 change "resistance” to “infection”
L173 change ”more resistance” to “less infection”
L181, 186,206, change “SEM” to “SD”
L283 “Rice seeds”, what cultivar, resistant or susceptible to AvrPik-D containing isolates?
L286 “benthamia” should be “benthamiana”
L352 [11]. Delete the period.
L354 add “after inoculation” after “10 days”
L365 confusing
L381 amount of “protease inhibitor cocktail”?
L388 “the acid”, what acid?
L407 change ”enhanced resistance to M. orzyae inoculation” to “reduced infection of M. oryzae”
Comments on the Quality of English LanguageThe quality of English language of this manuscript is good.
Author Response
Dear Editor and Reviewers,
Thanks very much for taking your time to review this manuscript. We really appreciate all your comments and suggestions! Please find my itemized responses in below and my revisions/corrections (color with red) in the re-submitted files.
Reviewer 3
Comments and Suggestions for Authors
Rice blast, caused by Magnaporthe oryzae, is a destructive disease of rice, threatening world food security. A series of experiments have been conducted in this research, and the results indicated that M. oryzae effector AvrPik-D interacts with Rubisco small unit OSRBCS4, overexpress OSRBCS4 could enhance ROS production and reduce infection. The findings could help understand the mechanisms of infection-defense interactions of this important disease. This manuscript may be publishable after revisions.
Major concerns
- Introduction didn’t sufficiently introduce the status of research on this subject. More details are needed, such as multiple alleles of rice Pik gene and variants of M. oryzae AvrPik have been identified. The Rubisco paragraph could be moved to “Discussion”. There is no need to describe the results in the Introduction part.
Response: This has been revised in the manuscript. Modify it on line 54-71, 237-253, and mark it in red.
2.Half of the discussion was a summary of the results, which is redundant and unnecessary since there is a Conclusion part. Different targets of M. oryzae Avr genes were identified, and the authors’ previous research found that AvrPik-D targets transcription factor WG7. The differences should be discussed.
Response: We have rewritten the discussion section and removed duplicate parts. The modified parts are marked in red.
3.The authors can cite their own paper or re-organize the language, especially the Materials and Methods part.
Response: This has been revised in the manuscript. The modified parts are marked in red.
Minor suggestions
L27 delete the 1st “global’
Response: This has been revised in the manuscript. Modify it on line 28 and mark it in red.
L45-46 delete “layer”
Response: This has been revised in the manuscript. Modify it on line 46 and mark it in red.
L88 “enhances resistance” should be replaced with “reduces infection”
Response: This has been revised in the manuscript. Modify it on line 92 and mark it in red.
L96 change “infection” to ‘infected”
Response: This has been revised in the manuscript. Modify it on line 96 and mark it in red.
L95-96 and L99-100, redundant and confusing.
Response: This has been revised in the manuscript. Modify it on L98-103 and mark it in red.
L103 change “respectively. And” to “respectively, and”
Response: This has been revised in the manuscript. Modify it on line 105 and mark it in red.
L106 change "To confirm that” to “To test if”
Response: This has been revised in the manuscript. Modify it on line 108 and mark it in red.
L107 change "fluorescence” to “fluorescent”
Response: This has been revised in the manuscript. Modify it on line 110 and mark it in red.
L166 change "resistance” to “infection”
Response: This has been revised in the manuscript. Modify it on line 168 and mark it in red.
L173 change“more resistance” to “less infection”
Response: This has been revised in the manuscript. Modify it on line 175 and mark it in red.
L181, 186,206, change “SEM” to “SD”
Response: This has been revised in the manuscript. Modify it on L183, 187, 207, and mark it in red.
L283 “Rice seeds”, what cultivar, resistant or susceptible to AvrPik-D containing isolates?
Response: MH86 rice is susceptible to strains containing AvrPikD. We add more information in the text. Modify it on line 290-291 and mark it in red.
L286 “benthamia” should be “benthamiana”
Response: This has been revised in the manuscript. Modify it on line 293 and mark it in red.
L352 [11]. Delete the period.?
Response: This has been revised in the manuscript. Modify it on line 374 and mark it in red.
L354 add “after inoculation” after “10 days”
Response: This has been revised in the manuscript. Modify it on line 376 and mark it in red.
L365 confusing
Response: This has been revised in the manuscript. Modify it on line 387-390 and mark it in red.
L381 amount of “protease inhibitor cocktail”?
Response: This has been revised in the manuscript. Modify it on line 408 and mark it in red.
L388 “the acid”, what acid?
Response: This acid is the termination reaction solution in the detection kit, and we are not sure what specific acid it is.
L407 change “enhanced resistance to M. orzyae inoculation” to “reduced infection of M. oryzae”
Response: We have made a change in the manuscript, which can be seen on line 433.